# On Success and Simplicity: A Second Look at Transferable Targeted Attacks

**Zhengyu Zhao** [1]  **Zhuoran Liu** [1]  **Martha Larson** [1]

## Abstract

Achieving transferability of targeted attacks is reputed to be remarkably difficult, and state-of-the-art approaches are resource-intensive due to training target-specific model(s) with additional data. In our work, we find, however, that simple transferable attacks which require neither additional data nor model training can achieve surprisingly high targeted transferability. This insight has been overlooked mainly due to the widespread practice of unreasonably restricting attack optimization to few iterations. In particular, we, for the first time, identify the state-of-the-art performance of a simple logit loss. Our investigation is conducted in a wide range of transfer settings, especially including three new, realistic settings: ensemble transfer with little model similarity, transfer to low-ranked target classes, and transfer to the real-world Google Cloud Vision API. Results in these new settings demonstrate that the commonly adopted, easy settings cannot fully reveal the actual properties of different attacks and may cause misleading comparisons. Overall, the aim of our analysis is to inspire a more meaningful evaluation on targeted transferability.

## 1. Introduction

Current work has achieved great success on non-targeted transferability (Dong et al., 2018; Zhou et al., 2018; Huang et al., 2019; Dong et al., 2019; Xie et al., 2019; Wu et al., 2020b; Li et al., 2020b; Lin et al., 2020; Gao et al., 2020), while several initial attempts (Liu et al., 2017; Dong et al., 2018; Inkawhich et al., 2019) on targeted transferability have shown its extreme difficulty. Targeted transferability is worth exploring since it can raise more practical concerns by causing a specific, highly dangerous prediction. However, so far state-of-the-art results can only be achieved by

resource-intensive transferable attacks (Inkawhich et al., 2020a;b; Naseer et al., 2021). Specifically, FDA (Inkawhich et al., 2020a;b) is based on modeling layer-wise feature distributions via class-specific auxiliary classifiers using large-scale labeled data, and then optimizing adversarial perturbations using these auxiliary classifiers from across the deep feature space. TTP (Naseer et al., 2021) is based on training target-specific Generative Adversarial Networks (GANs) through global and local distribution matching, and then use the trained generator to directly generate perturbations on any given input image.

In this paper, we take a second, thorough look at current research on targeted transferability. Our main contribution is the finding that *simple transferable attacks*, which require neither additional data nor model training but only using simple transfer techniques (MI (Dong et al., 2018), TI (Dong et al., 2019), and DI (Xie et al., 2019)), can achieve surprisingly high targeted transferability. We argue that this insight has been overlooked mainly because current research has unreasonably restricted attack optimization to a limited number of iterations. Another key contribution of our work is, for the first time, identifying the superiority of a simple logit loss, which even achieves new state-of-the-art results. A detailed review of related work on transferable targeted attacks can be found in Appendix A.

We demonstrate the general effectiveness of simple transferable attacks in a wide range of transfer settings, especially including three new, realistic settings: ensemble transfer with little model similarity, transfer to low-ranked target classes, and transfer to the real-world Google Cloud Vision API. These new settings can better reveal the actual properties of different attacks than the commonly adopted, easy settings. Overall, this paper elucidates the weakness in common practice and the limitations of the commonly adopted transfer settings. We hope our analysis will inspire a more meaningful evaluation on targeted transferability.

## 2. New Insights into Simple Transferable Attacks

In this section, we revisit simple transferable attacks in the targeted scenario, and provide new insights into them.

**Simple transferable attacks are surprisingly good when**

---

[1]Institute for Computing and Information Sciences, Radboud University, The Netherlands. Correspondence to: Zhengyu Zhao <z.zhao@cs.ru.nl>.

*Accepted by the ICML 2021 workshop on A Blessing in Disguise: The Prospects and Perils of Adversarial Machine Learning.* Copyright 2021 by the author(s).

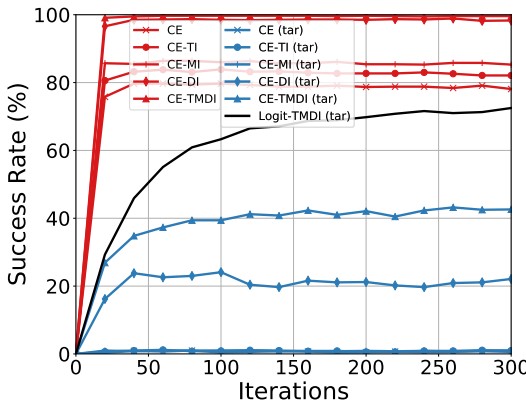

*Figure 1.* Transfer success rates of simple transferable attacks using CE or logit loss in the non-targeted and targeted scenarios.

**given enough iterations to converge.** As can be seen from Figure 1, while using TI or MI makes little difference to the original poor targeted transferability, somewhat surprisingly, using DI actually yields substantial targeted transferability, and integrating all three transfer techniques leads to the best performance. This may be explained by the fact that DI can yield more generalizable gradients by randomizing the image augmentation parameters not only across the image dimension but also across iterations. However, current work has missed this point since most studies have been focused on the MI technique (Liu et al., 2017; Dong et al., 2018; Inkawhich et al., 2019; 2020a;b).

We can also observe that the simple transferable attacks can converge to much higher targeted transferability with more iterations, in contrast to the fast success in the non-targeted scenario. Current work has not realized this difference and so naturally followed the research on non-targeted transferability to use few iterations (typically $\leq 20$) also for targeted transferability. However, evaluation with few iterations is definitely problematic. On the one hand, comparing different optimization processes that have not converged is not meaningful and may cause misleading results. This observation is consistent with Carlini et al. (2019), who pointed out that restricting the number of iterations without verifying the attack convergence is one of the common pitfalls in evaluating adversarial robustness. Several recent defenses have been defeated by simply using more iterations (Tramer et al., 2020). On the other hand, considering the realistic threat model, it is not meaningful to artificially restrict the computational power of a practical attack (e.g., to fewer than several thousand attack iterations) (Athalye et al., 2018).

**A simple yet stronger logit attack.** Existing simple transferable attacks are commonly built up on the cross-entropy loss. However, as pointed out by Li et al. (2020a), the cross-entropy loss may suffer from the vanishing gradient problem. In our case with many iterations, this problem

becomes more serious since it will cause the attack to stop improving at some point. To address this problem, we adopt a straightforward idea by directly maximizing the target logit for continuously pushing the image deep into the territory of the target class. This simple logit loss can be expressed as $L_{Logit} = -l_t(\boldsymbol{x}')$, where $l_t(\cdot)$ denotes the logit output with respect to the target class. Although this logit loss is very similar to the well-known C&W loss (Carlini & Wagner, 2017), its superiority on targeted transferability has not been recognized so far. We find that directly using the C&W loss performs worse than this logit loss. For example, in the single-model transfer setting with the ResNet50 as the white-box source model, the C&W loss yields 20% worse performance on average. It may be because the C&W loss also involves suppressing other classes, which is not necessary here but could trade off the high target logit. As can be seen from Figure 1, this logit loss consistently yields better performance than the commonly used cross-entropy loss, and the performance gap becomes increasingly larger as the number of iterations rises.

# 3. Experimental Evidence on Simple Transferable Attacks

In this section, we provide experimental evidence to show the effectiveness of simple transferable attacks. Firstly, we evaluate them in a variety of transfer settings, including single-model transfer, ensemble transfer (easy and challenging settings), a worse-case setting with low-ranked target classes, and a real-world attack against the Google Cloud Vision API. Then, we compare them with two state-of-the-art resource-intensive transferable attacks, TTP (Naseer et al., 2021) and FDA (Inkawhich et al., 2020b).

We use the 1000 images from the development set of ImageNet-Compatible Dataset (Kurakin et al., 2018), and all these images are associated with 1000 ImageNet class labels and cropped to $299 \times 299$ before use. We consider four diverse classifier architectures: ResNet, DenseNet, VGGNet, and Inception. Our experiments are run on a single NVIDIA Tesla P100 GPU with 12GB of memory.

We test three different attack losses: cross-entropy (CE), Po+Trip and logit. For each image, we use the target label that was officially specified in the dataset. We use a moderate step size of 2 for all attacks, and find that the performance is not sensitive to step size (see evidence in Appendix C). Following the common practice, the perturbations are restricted by $L_\infty = 16/255$. If not mentioned specifically, 300 iterations are used for each attack. All attacks use TI, MI, and DI together with their original optimal hyperparameters. Specifically, $\|\boldsymbol{W}\|_1 = 5$ is used for 'TI' as suggested by (Gao et al., 2020). When being executed with a batch size of 20, the optimization of each attack takes about three second per image.

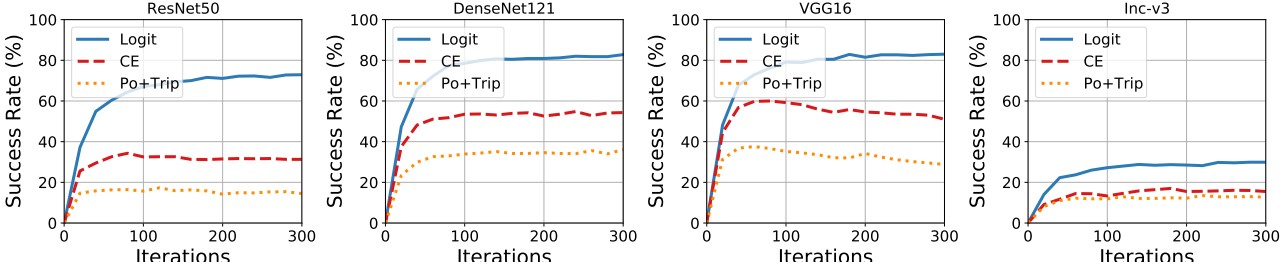

*Figure 2.* Targeted success rates (%) for our realistic ensemble transfer with little model similarity.

*Table 1.* Targeted success rates (%) for single-model transfer.

| Attack | Source Model: R50 | | | Source Model: D121 | | |
|--------|------|------|-------|------|------|-------|
| | →D121 | →V16 | →Inv3 | →R50 | →V16 | →Inv3 |
| CE | 42.6 | 30.4 | 4.1 | 19.4 | 10.9 | 3.5 |
| Po+Trip | 54.7 | 34.4 | 5.9 | 14.7 | 7.7 | 2.7 |
| Logit | **72.5** | **62.7** | **9.4** | **43.7** | **38.7** | **7.6** |

| Attack | Source Model: V16 | | | Source Model: Inv3 | | |
|--------|------|-------|-------|------|-------|------|
| | →R50 | →D121 | →Inv3 | →R50 | →D121 | →V16 |
| CE | 0.6 | 0.1 | 0 | 2.4 | 2.9 | 2.0 |
| Po+Trip | 0.5 | 0.7 | 0.1 | 2.5 | 3.3 | 2.0 |
| Logit | **11.2** | **13.2** | **0.9** | **2.9** | **5.3** | **3.7** |

*Table 2.* Targeted success rates (%) for transfer with varied targets.

| Attack | 2nd | 10th | 200th | 500th | 800th | 1000th |
|--------|------|------|-------|-------|-------|--------|
| CE | **89.9** | 76.7 | 49.7 | 43.1 | 37.0 | 25.1 |
| Po+Trip | 82.6 | 77.6 | 58.4 | 53.6 | 49.1 | 38.2 |
| Logit | 83.8 | **81.3** | **75.0** | **71.0** | **65.1** | **52.8** |

**Single-model transfer.** Table 1 reports the targeted transferability in cases with different classifiers. As can be seen, the logit loss outperforms the other two by a large margin in all cases. When comparing different model architectures, we can find that the attacks generally achieved lower performance when transferring from the VGGNet16 or Inception-v3 than from ResNet50 or DenseNet121. This is consistent with the observations in (Inkawhich et al., 2020a;b), and may be explained by the fact that skip connections (in ResNet50 and DenseNet121) allow easier generation of highly transferable adversarial examples (Wu et al., 2020a). The very low performance when transferring to Inception-v3 might be explained by its notable difference with the other three architectures, i.e., the Inception architecture is known to be more conceptually complex (heavily engineered) with multiple-size convolution and two auxiliary classifiers.

**Ensemble transfer with little model similarity.** A common approach to boosting transferability is to transfer from an ensemble of multiple source models. Following the common practice, we simply assign equal weights to all the source models. We found that the commonly adopted ensemble settings (Dong et al., 2018; Tramèr et al., 2018; Dong et al., 2019; Li et al., 2020a) that only involve similar model architectures are so easy that all attacks can reach very high targeted transferability (see evidence in Appendix B).

Therefore, in order to fully reveal the potential of different attacks, we consider a more challenging transfer setting with no architectural overlap between the source ensemble and the target model. This setting is also more realistic since in the real-world attack scenarios, there is almost unlikely to be a local white-box source model that shares similar

architecture to the unknown target model. Figure 2 shows that the logit attack largely outperforms the other two attacks in this realistic ensemble setting. Po+Trip yields even worse results than the CE loss maybe because its original hyperparameters cannot ensure optimal effectiveness here. Again, Inception-v3 is the most difficult model to attack.

**A worse-case transfer setting with low-ranked target classes.** In conventional security studies, a comprehensive evaluation of attacks commonly involves a range of settings with varied difficulties. For adversarial attacks in the white-box scenario, existing studies (Carlini & Wagner, 2017; Kurakin et al., 2017; Tramèr et al., 2018; Rony et al., 2019) have also looked at varied difficulties regarding the target class. Specifically, in the best case, the targeted success is basically equal to non-targeted success, i.e., an attack is regarded to be successful as long as it can success on any arbitrary target other than the original class. In the average case, the target class is randomly specified agnostic to the test image, while in the worst case, the target is specified as the lowest-ranked (least-likely) class in the prediction list of the original image.

However, to the best of our knowledge, current evaluation on transferability has been limited to the best and average cases. To address this limitation, we consider a worse-case transfer setting by varying the target from the highest-ranked class gradually to the lowest one. As can be seen from Table 2, there exists non-negligible correlation between the ranking position of the target class and the transferability. More specifically, it becomes increasingly difficult as the target moves down the prediction list. We can also observe that only looking at the results with higher-ranked targets could not reveal the actual properties of different attacks and may lead to misleading conclusion.

**Transfer-based attacks on Google Cloud Vision.** Exist-

*Table 3.* Success rates (%) for transfer to Google Cloud Vision. An ensemble of all four diverse models is used as the source model.

|              | CE | Po+Trip | Logit |
|--------------|----|---------|-------|
| Targeted     | 7  | 8       | **18** |
| Non-targeted | **51** | 44  | **51** |

*Table 4.* Targeted success rates of Simple vs. TTP. D121 or V16 is used as the target model, and R50 or an ensemble (-ens) of R{18,50,101,152} is the source model.

| Attack | D121 | V16 | D121-ens | V16-ens | Average |
|--------|------|-----|----------|---------|---------|
| TTP    | **79.6** | **78.6** | 92.9 | 89.6 | 85.2 |
| Logit  | 75.9 | 72.5 | **99.4** | **97.7** | **86.4** |

ing work that has studied transfer-based attacks against real-world systems were only explored in the non-targeted scenario, and limited to face recognition (Shan et al., 2020; Cherepanova et al., 2021; Rajabi et al., 2021). In contrast, we consider a more generally-used image recognition system, the Google Cloud Vision API, in a more challenging, targeted scenario. For targeted transferability, we evaluate whether or not the target object class appears among the returned (top 10) predictions. We also report the non-targeted results to give extra insight by evaluating whether or not the correct object class appears among the returned predictions. Since the Google Cloud Vision predictions do not explicitly correspond to the 1000 ImageNet classes, we treat semantically similar classes as the same class.

Table 3 reports the results averaged over 100 images that originally yield correct predictions. As expected, targeted transfer is strictly more difficult than the non-targeted transfer. The logit attack achieved the best results, especially in the targeted scenario. Figure 4 in Appendix D visualizes some adversarial images, which are shown to have quasi-imperceptible perturbations. In general, our results reveal the potential vulnerability of Google Cloud Vision against simple attacks that need no query interaction.

**Compared with resource-intensive transferable attacks.** We firstly compare the best performed simple attack, the logit attack, with the state-of-the-art TTP, following its original "10-Targets (all-source)" setting (Naseer et al., 2021). As shown in Table 4, the logit attack achieved high targeted transferability comparable to TTP in both transfer settings, with slightly better performance on average.

We next compare all three simple transferable attacks with FDA (Inkawhich et al., 2020b). We chose only the setting with "distal transfer" examples since the authors have not (yet) released their source code. Specifically, the adversarial examples are generated by starting from random Gaussian noise and optimizing without any perturbation bounds to unleash the full potential of the attacks. The results are

*Table 5.* Targeted success rates of Simple vs. FDA for unbounded transfer. All attacks use 200 iterations.

|          | FDA | CE | Po+Trip | Logit |
|----------|-----|-----|---------|-------|
| R50→D121 | 65.8 | 69.3 | **88.1** | 84.1 |
| R50→V16  | 48.1 | 54.1 | 67.8 | **74.2** |
| Average  | 57.0 | 61.7 | 78.0 | **79.2** |

averaged over 4000 examples, each of which is optimized towards a random target class. As suggested by (Inkawhich et al., 2020b), the MI technique is removed since it empirically harms the performance in this unbounded case. Table 5 shows that all three simple transferable attacks achieved higher targeted transferability than FDA, with the logit loss achieving the best overall performance. As can be seen from Figure 5 in Appendix E, the "distal transfer" examples can somehow reflect the target semantics. This suggests an interesting perspective that targeted transferability is achieved by attacking the robust features because these features are naturally shared by different models (and also humans). This finding is different from previous perspective on non-targeted transferability, for which attacking the non-robust features is sufficient (Ilyas et al., 2019).

## 4. Conclusion and Outlook

In this paper, we have found that simple transferable attacks can achieve surprisingly high transferability as long as they are not unreasonably restricted to few iterations. We have validated the effectiveness of simple transferable attacks in a wide range of transfer settings, including three newly-introduced realistic settings that better revealed the actual properties of different attacks. In particular, we identify that a very simple logit attack can consistently yield the highest targeted transferability, being even competitive with the state-of-the-art resource-intensive approaches. Overall, we hope our findings on weakness in common practice and the limitations of the commonly adopted transfer settings will inspire future research to conduct a more meaningful evaluation on targeted transferability.

We hope our analysis can motivate the community to design stronger defenses against transferable attacks, and on the other hand, promote the applications that directly use adversarial examples for social good, such as protecting privacy (Oh et al., 2017; Larson et al., 2018; Liu et al., 2019; Cherepanova et al., 2021; Rajabi et al., 2021). For future work, it is worth a deeper understanding of why different model architectures yield different transferability.

## Acknowledgments

This work was carried out on the Dutch national e-infrastructure with the support of SURF Cooperative.

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

## A. Related Work on Transferable Attacks

In this section, we review simple transferable attacks and recently-proposed resource-intensive transferable attacks.

**Simple transferable attacks** that require neither additional data nor model training have been extensively studied in the non-targeted scenario (Dong et al., 2018; Zhou et al., 2018; Dong et al., 2019; Huang et al., 2019; Xie et al., 2019; Lin et al., 2020; Li et al., 2020b; Wu et al., 2020b; Gao et al., 2020), and also explored in the targeted scenario (Liu et al., 2017; Dong et al., 2018; Li et al., 2020a). These attacks are commonly built up on the well-known Iterative-Fast Gradient Sign Method (I-FGSM) (Kurakin et al., 2017; Madry et al., 2018). In the targeted scenario, I-FGSM can be formulated as:

$$x_0' = x, \quad x_{i+1}' = x_i' - \alpha \cdot \text{sign}(\nabla_x J(x_i', y_t)), \quad (1)$$

where $x_i'$ denotes the intermediate modified image in the $i$-th iteration, and $y_t$ is the target class label. In order to ensure the imperceptibility of the modifications, in each iteration, the perturbations are clipped into some pre-defined $L_p$ norm ball, i.e., satisfying $\|x' - x\|_p \leq \epsilon$. Current research on transferable attacks has commonly adopted the $L_\infty$ norm. For the loss function $J(\cdot, \cdot)$, most simple transferable attacks have adopted the plain cross-entropy (CE) loss.

However, this commonly used CE loss has been recently pointed out to be not effective in the targeted scenario because it suffers from the vanishing gradient problem (Li et al., 2020a). Therefore, the authors in (Li et al., 2020a) have proposed a so-called Po+Trip loss to address the problem. Specifically, the Poincaré distance was used to adapt the magnitude of the gradients, which can be formulated as:

$$L_{Po} = d(u, v) = \text{arccosh}(1 + \delta(u, v)),$$
$$\delta(u, v) = \frac{2 \cdot \|u - v\|_2^2}{(1 - \|u\|_2^2)(1 - \|v\|_2^2)}, \quad (2)$$

where $u$ is the normalized logit vector and $v$ is the one-hot vector with respect to the target class. In order to further boost the performance, an additional triplet loss was inte-

grated to also push the image away from the original class:

$$L_{Trip} = [D(l(x'), y_t) - D(l(x'), y_o) + \gamma]_+,$$
$$D(l(x'), y) = 1 - \frac{\|l(x') \cdot y\|_1}{\|l(x')\|_2 \|y\|_2}. \quad (3)$$

The overall loss function can be formulated as $L_{Po+Trip} = L_{Po} + \lambda L_{Trip}$. However, in the original work, Po+Trip was evaluated only in an ensemble transfer setting.

Instead of improving the loss function, there are also transfer techniques (Dong et al., 2018; 2019; Xie et al., 2019; Lin et al., 2020) that aim at preventing the attack optimization from overfitting to the specific source model. The advantage of such techniques is that they can be generally applied to any loss functions. In this paper, we study three (Dong et al., 2018; 2019; Xie et al., 2019) of such transfer techniques that have been widely used in the literature:

Momentum Iterative-FGSM (MI-FGSM) (Dong et al., 2018) integrates a momentum term, which accumulates previous gradients in order to make more accurate updating. It can be expressed as:

$$g_{i+1} = \mu \cdot g_i + \frac{\nabla_x J(x_i', y_t)}{\|\nabla_x J(x_i', y_t)\|_1},$$
$$x_{i+1}' = x_i' - \alpha \cdot \text{sign}(g_i), \quad (4)$$

where $g_i$ is the accumulated gradients at the $i$-th iteration, and $\mu$ is the decay factor. Another similar technique that is based on the Nesterov accelerated gradient was also explored (Lin et al., 2020).

Translation Invariant-FGSM (TI-FGSM) (Dong et al., 2019) randomly translates the input image during optimization for preventing the attack from overfitting to the specific white-box source model. This approach is inspired by the common data augmentation techniques used for preventing overfitting in model training. Instead of calculating gradients for multiple translated images separately, the authors proposed an approximate solution to accelerate the implementation. It is achieved by directly computing locally smoothed gradients on the original image via convolution with a kernel:

$$x_{i+1}' = x_i' - \alpha \cdot \text{sign}(W * \nabla_x J(x_i', y_t)), \quad (5)$$

where $W$ is the convolution kernel used for smoothing. TI-FGSM was originally designed for transferring to adversarially-trained models. It has been recently pointed out that using relatively small kernel size leads to the optimal transferability when transferring to normally trained models (Gao et al., 2020).

Diverse Input-FGSM (DI-FGSM) (Xie et al., 2019) is conceptually similar to TI-FGSM, but applies random resizing and padding for the image augmentation. More importantly, instead of fixing the augmentation parameters all the time

*Table 6.* Targeted success rates (%) in the commonly adopted, easy ensemble transfer setting, where the hold-out target model (denoted by '-') and the ensemble models share similar architectures. Results with 20/100 iterations are reported.

| Attack | -Inc-v3 | -Inc-v4 | -IncRes-v2 | -Res50 | -Res101 | -Res152 | Average |
|---|---|---|---|---|---|---|---|
| CE | 48.8/85.3 | 47.2/83.3 | 47.5/83.9 | 50.9/89.8 | 58.5/**93.2** | 56.7/90.7 | 51.6/87.7 |
| Po+Trip | **59.3**/84.4 | **55.0**/82.4 | 51.4/80.8 | 56.9/85.0 | 60.5/87.9 | 57.6/85.7 | 56.8/84.4 |
| Logit | 56.4/**85.5** | 52.9/**85.8** | **54.4**/**85.1** | **57.5**/**90.0** | **64.4**/91.4 | **61.3**/**90.8** | **57.8**/**88.1** |

as in TI-FGSM, DI-FGSM adds more randomness across iterations. The attack optimization can be formulated as:

$$\boldsymbol{x}'_{i+1} = \boldsymbol{x}'_i - \alpha \cdot \text{sign}(\nabla_{\boldsymbol{x}} J(T(\boldsymbol{x}'_i, p), y_t)), \quad (6)$$

where the stochastic transformation $T(\boldsymbol{x}'_i, p)$ is implemented with probability $p$ at each iteration.

**Resource-intensive transferable attacks** that require training target-specific models on large-scale additional data have been recently explored and shown to substantially improve targeted transferability. Specifically, Feature Distribution Attack (FDA) (Inkawhich et al., 2020a) has substantially improved targeted transferability on ImageNet by training auxiliary models with additional labeled data. Each auxiliary model is a small, binary, one-versus-all classifier trained for a specific target class at a specific layer. That is to say, the number of auxiliary models is the number of layers probed multiplied by the number of target classes that are required to model (Inkawhich et al., 2020a).

The attack loss function can be formulated as:

$$L_{FDA} = J(\mathcal{F}_l(\boldsymbol{x}'), y_t) - \eta \frac{\|\mathcal{F}_l(\boldsymbol{x}') - \mathcal{F}_l(\boldsymbol{x})\|_2}{\|\mathcal{F}_l(\boldsymbol{x})\|_2}, \quad (7)$$

where each auxiliary models $F_l(\cdot)$ can model the probability that a feature map at layer $l$ is from a specific target class $y_t$. FDA$^{(N)}$+xent (Inkawhich et al., 2020b) extends FDA by aggregating features from $N$ layers and also incorporating the cross-entropy loss $H(\cdot, \cdot)$ of the original network $\mathcal{F}(\cdot)$. The extended loss function can be expressed as:

$$L_{FDA^{(N)}+xent} = \sum_{l \in L} \lambda_l(L_{FDA} + \gamma H(\mathcal{F}(\boldsymbol{x}'), y_t)),$$
$$\text{where} \sum_{l \in L} \lambda_l = 1. \quad (8)$$

Very recently, TTP (Naseer et al., 2021) has achieved state-of-the-art targeted transferability by directly generating perturbations using target class-specific GANs that have been trained with global and local distribution matching. Specifically, the global distribution matching is achieved by minimizing the Kullback Leibler (KL) divergence, while the local distribution matching is by enforcing the neighbourhood similarity. In order to further boost the performance, image augmentation techniques, such as rotation, crop resize,

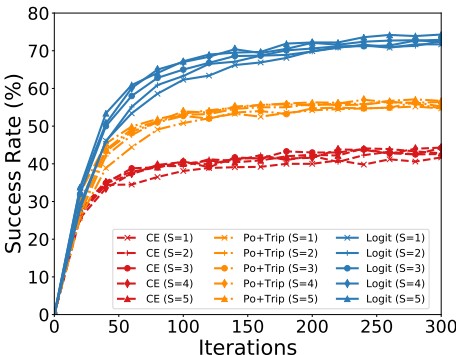

*Figure 3.* Targeted success rates with varied step sizes, S.

horizontal flip, color jittering and gray scale transformation, have been applied during training. We refer the readers to the original work for more technical details of TTP.

## B. Results in the Easy Ensemble Setting

Here we look at the commonly adopted ensemble transfer setting (Dong et al., 2018; 2019; Li et al., 2020a; Tramèr et al., 2018) in which each hold-out target model shares similar architecture with some of the white-box ensemble models. As can be seen from Table 6, all the three attacks reached equally high targeted transferability when given enough iterations to converge. This indicates that this setting with high model similarity could not fully reveal the actual properties of different attacks. We can also observe that Po+Trip performs better than the CE loss only at 20 iterations, but becomes worse when they are given enough iterations to converge. This finding suggests that evaluating different attacks under only few iterations may cause misleading comparing results.

## C. Results with Varied Step Sizes

Recent work has shown that enlarging the step size can improve non-targeted transferability since it can help attack optimization escape from poor local optima (Gao et al., 2020). Here we also explore the impact of step size setting on targeted transferability. As can be seen from Figure 3, in general, all attacks are not sensitive to the change of step size, with only a slight improvement when using a larger step size. We can also observe that the logit attack consistently outperforms the other two in all cases.

## D. Adversarial Images on Attacking Google Cloud Vision

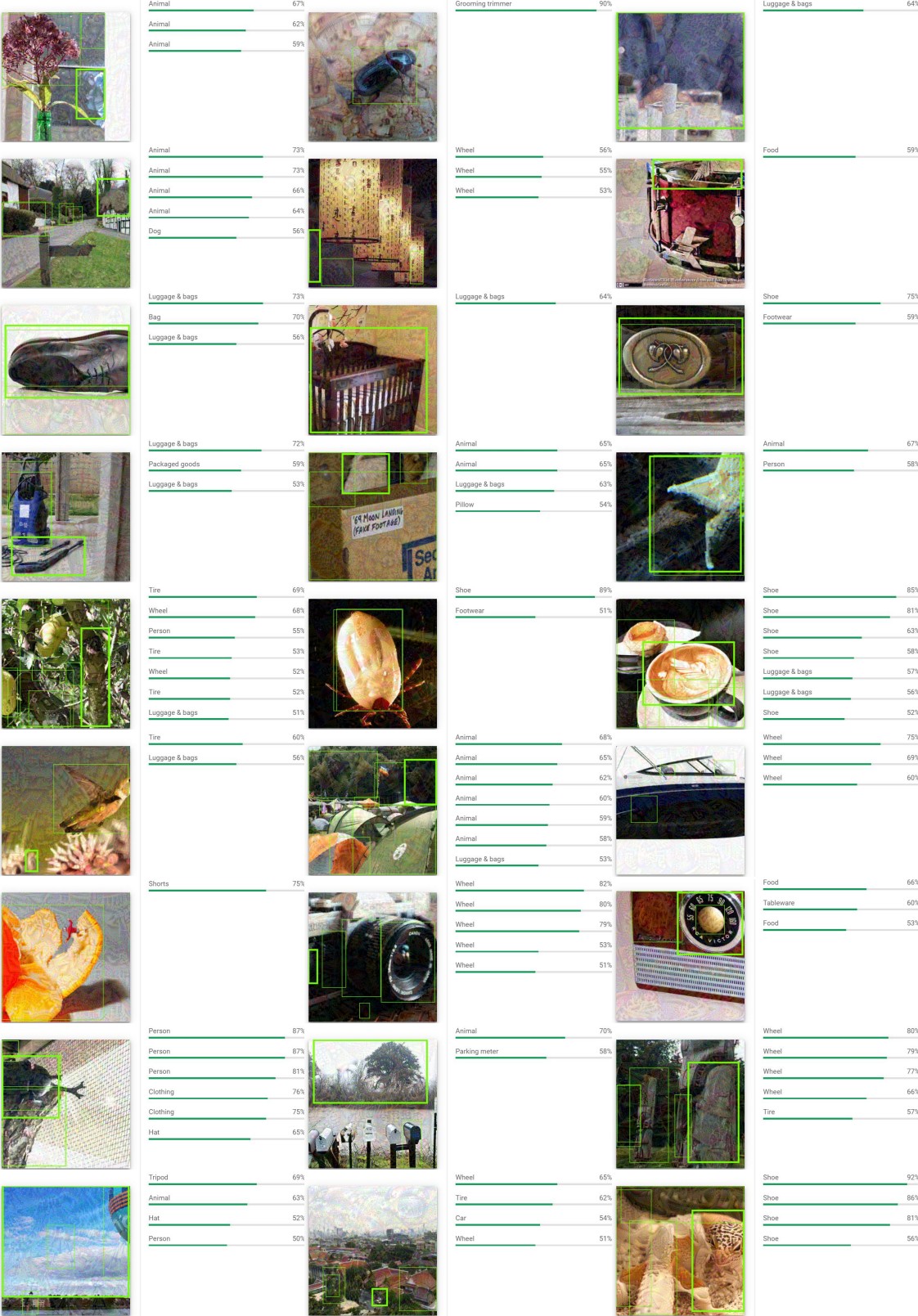

*Figure 4.* Adversarial images on attacking Google Cloud Vision.

# E. "Distal Transfer" Examples with Unbounded Perturbations

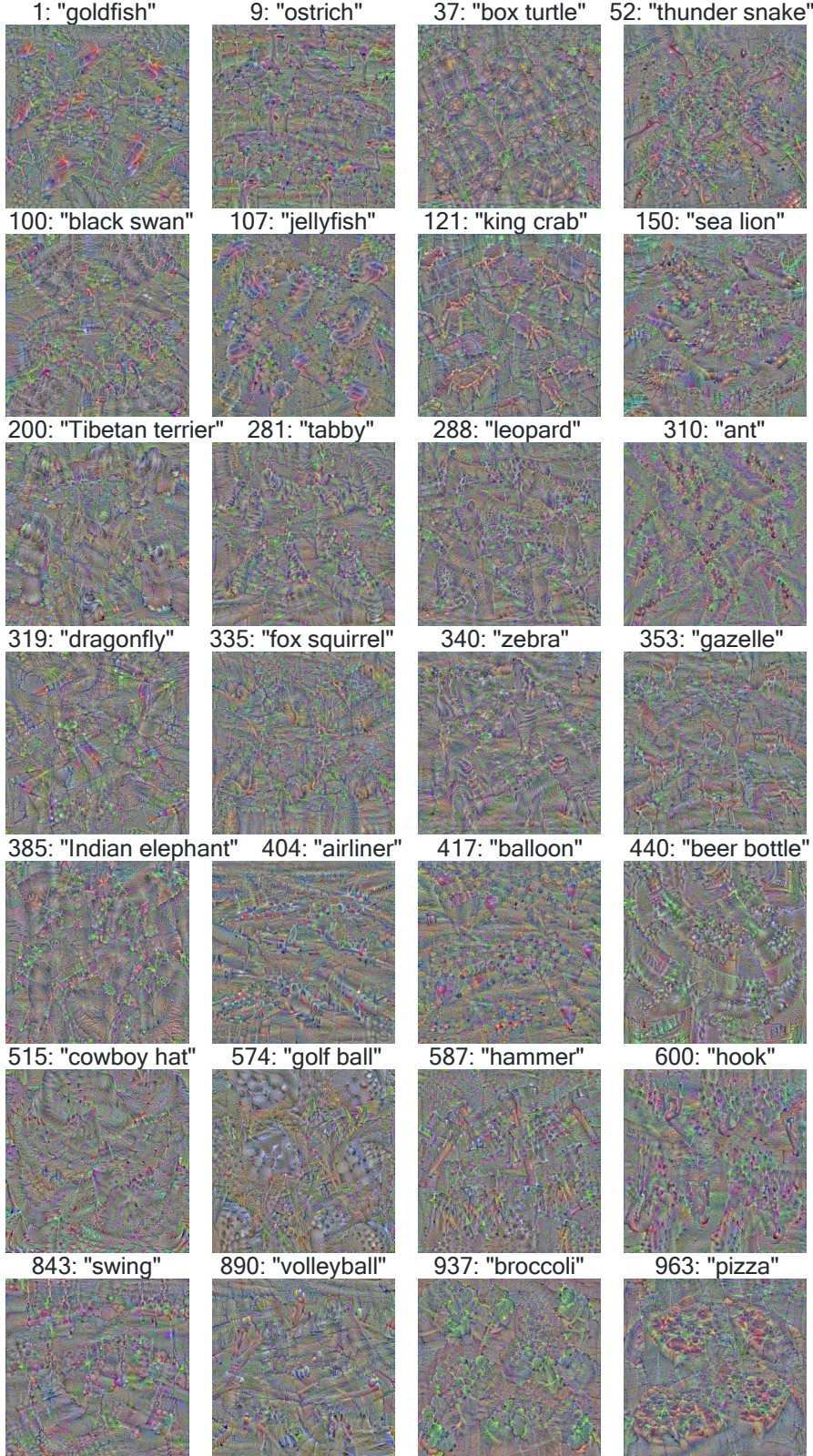

*Figure 5.* "Distal transfer" examples with unbounded perturbations.