# OpenReview forum: "On Success and Simplicity: A Second Look at Transferable Targeted Attacks"
_ICML.cc/2021/Workshop/AML — ICML 2021 Workshop AML Poster_

### Official Review · Reviewer_3qKJ · 2021-06-19
**This paper is easy to follow. However, the paper is in lack of novelty.**

**Rating:** Accept
**Confidence:** 4

**Review:**

pros:
1. This paper is easy to follow, and the structure is complete.
2. The proposed logit attack is simple but effective in targeted adversarial attack.
cons:
1. The word 'form' between line 074 and 075 in the left column should be 'from'.
2. The experiment results are not so convincing. It is claimed that directly using the C&W loss performs worse than this logit loss. However, the comparison experiment with C&W loss is not mentioned. Actually, there exists targeted C&W loss.
3. The design of logit loss is actually too simple and included in C&W loss. It is in lack of novelty.

---

### Decision · Program_Chairs · 2021-06-21

**Decision:**

Accept (Poster)

**Comment:**

The reviewer noticed several advantages of this paper as well as some shortcoming. The authors can further address the reviewer's comments.